# Recurrent Embedded Topic Model

Carlos Vargas [1,*,†] and Hiram Ponce [2,†]

[1]  Facultad de Ingeniería, Universidad Panamericana, Josemaría Escrivá de Balaguer 101, Aguascalientes 20290, Aguascalientes, Mexico
[2]  Facultad de Ingeniería, Universidad Panamericana, Augusto Rodin 498, Mexico City 03920, Mexico; hponce@up.edu.mx
*   Correspondence: 0250154@up.edu.mx
†   These authors contributed equally to this work.

**Abstract:** In this paper we propose the Recurrent Embedded Topic Model (RETM) which is a modification of the Embedded Topic Modelling (ETM) by reusing the Continuous Bag of Words (CBOW) that the model had implemented and applying it to a recurrent neural network (LSTM), using the order of the words of the text, in the CBOW space as the recurrency of the LSTM, while calculating the topic–document distribution of the model. This approach is novel because the ETM and Latent Dirichlet Allocation (LDA) do not use the order of the words while calculating the topic proportions for each text, making worse predictions in the end. The RETM is a topic-modelling technique that vastly improves (by more than 15 times in train data and between 10% and 90% better based on test dataset values for perplexity) the quality of the topics that were calculated for the datasets used in this paper. This model is explained in detail throughout the paper and presents results on different use cases on how the model performs against ETM and LDA. The RETM can be used with better accuracy for any topic model-related problem.

**Keywords:** topic modelling; natural language processing; recurrent embedded topic model; latent dirichlet allocation; embedded topic model

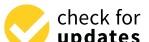

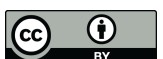

## 1. Introduction

In recent years, understanding what is stored inside big data has proved to be a challenge. According to the estimations of Petroc Taylor in Statista [1], in 2023, there should be around 120 zettabytes of data created, captured, consumed and copied worldwide. This trend keeps getting higher every single year (Table 1). Experts from Intersystems [2] estimate that 85% of all data that exist are unstructured data, most of which is held in text-based documents such as emails, memos, documents, social media feeds, etc.

**Table 1.** Table shows an estimation of how many zettabytes are currently stored worldwide. Data from Petroc Taylor in Statista [1].

| Year | Number of Zettabytes Estimated to Be Stored Worldwide |
| --- | --- |
| 2020 | 64.2 |
| 2021 | 79 |
| 2022 | 97 |
| 2023 | 120 |
| 2024 | 147 |
| 2025 | 181 |

Topic modelling is an area of data analytics that analyzes the words of texts to discover the themes that run through them and help with the organization of unstructured, text-based data at a scale that would be humanly impossible [3].

Several topic models exist but there has been a clear evolution in the quality of the topic models over the years. Some of the common ones are based on latent semantic analysis but were outperformed by Latent Dirichlet Allocation (LDA), which is also currently one of the most popular models [4,5].

LDA has been used in a variety of fields and for a variety of use cases. Some of the highlights are in political science (where 50 years' worth of data was fitted into LDA to descriptively understand the solutions and strategies that have been adopted to reduce crime) [6], software engineering (where LDA was fitted on source code from 1555 projects and managed to extract the main concepts from the source code) [7] and social media (where a modification of LDA was used to align events and Twitter feedback) [8].

Although LDA performs a good job at topic modelling, when using specific metrics to measure quality such as perplexity, it still has a long way to go. Perplexity is a commonly used metric for measuring the performance of language models [9]. When LDA is measured by perplexity against modifications of the model, it proves to be outperformed every time [10–12], demonstrating that there is still room for improvement in modifying LDA.

LDA assumes that each document can be represented by a mixture of topics. The aim of the model is to create a distribution with the proportions of how much every topic has been mentioned, throughout the document, and a distribution with the importance of each word on representing each topic. When LDA is fitted to a database of documents, it will provide a low-dimensional representation for each document and word representations for each topic [5].

On the other hand, there have been realizations in other unsupervised learning problems, such as transforming raw words into vectors. The embedding layer was first proposed by Bengio et al. [13] and was later explored by Mikolov et al. [14,15], where they noticed a high level of accuracy for words that have similar syntactic and semantic relationships [14] and can also capture synonyms, antonyms and spelling variations [16], until finally arriving at the Continuous Bag of Words (CBOW) embedding layer [15].

Both models (LDA and CBOW) have been used together by Dieng et al. [17] in their embedded topic modelling (ETM). The ETM can be seen as the best of its two parts: from one side, as a topic model, it can provide a low-dimensional vector with the proportions of the topics inside the text trying to generate all the syntactic and semantic relationships; on the other side, it acts as an embedding model, where it can represent the vocabulary as more robust and standardized vectors and exploit the relationships between words, making the ETM a superior topic model. An important weakness that LDA tends to demonstrate is based on the size of the vocabulary, where the bigger the vocabulary, the lower the accuracy (measured by the Coherence-Normalized Perplexity metric); thus, one of the biggest strengths of the ETM is making more accurate predictions where the vocabulary is bigger, as shown by Figure 1. It is important to note that even though when the size of the vocabulary is small, the ETM still outperforms LDA.

Although the ETM works great as a combination of LDA and CBOW, it is unable to analyze the order of the words in the text. If the topic models were able to use the sequence of the words, they would outperform current methodologies for calculating topic proportions. To use the order of the words in the text, this issue can be tackled with the recurrency of a Recurrent Neural Networks (RNN). To be specific, it can be tackled by the use of Long Short-Term Memory neural networks (LSTMs) [18], which have been experimented on, with great success, in Natural Language Processing (NLP) problems [19].

In this paper, we propose a new modification to the current ETM (Recurrent Embedded Topic Modelling, RETM) by implementing LSTMs into the NN and making adjustments to the overall architecture, where, measured by the perplexity metric, it severely outperforms the ETM and LDA. The intuition behind using an LSTM is to remove the possibility of having the vanishing/exploding gradient problem that usually hurts recurrent neural networks (given that the recurrency for this problem is based on the order of the words). Some of the experiments that were used to validate the declaration of the RETM outperforming LDA and ETM are based on four different datasets, containing text-based data from emails,

books, news and movies which were utilized for calculating the topic distributions. In all four datasets RETM proved to have an improvement of at least 15 times better in the training dataset and at least 10% improvement in test dataset.

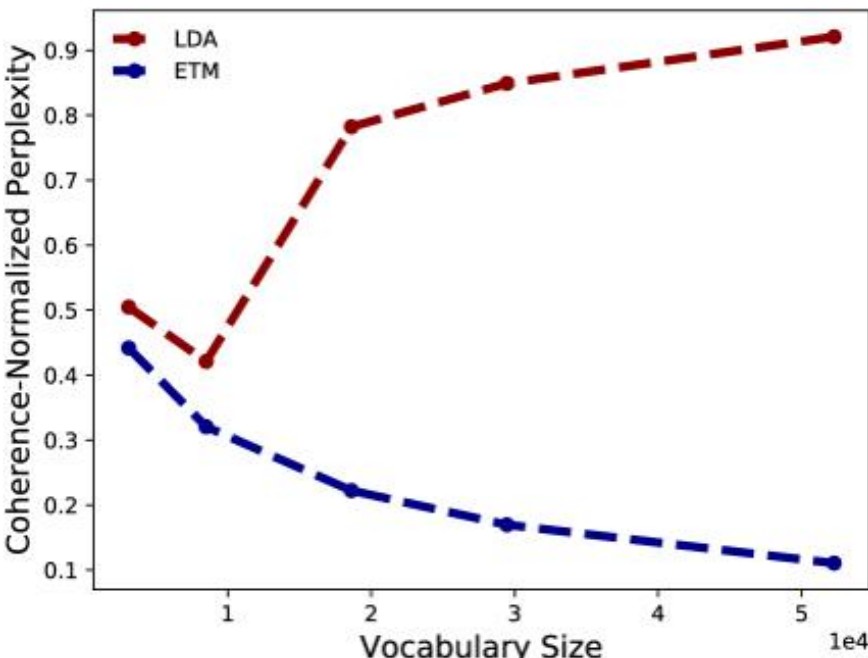

**Figure 1.** Comparison between the ETM and LDA using the coherence-normalized perplexity metric over different vocabulary sizes [17].

In a nutshell, the reasoning behind the modification was to use the word order of the documents and reuse the embedding layer from the CBOW model to generate the values of each word inside the calculation of the topic–document matrix (theta) from the ETM model and thus improve the topic assignment for each text.

The contribution of this work is based on the perplexity metric for the topic models: when the ETM is fitted with an LSTM using the word order and the embedding layer is used as the recurrency for the RNN while calculating the topic–document distribution, an improvement of at least 15 times on average in the training dataset for all models tested (Combined Topic Model, ZeroShot Cross-lingual Topic Model, LDA, ETM) and an improvement of at least 10% over test data were achieved, ensuring higher-quality topics and making the models more successful for topic modelling-related tasks.

To the best of our knowledge, this is the first time recurrence has been implemented in a topic model, as we propose with RETM.

The rest of the paper is organized as follows: Section 2 tackles the related work; later, in Section 3, the mathematical support of the model can be found alongside the inference and estimation and the pseudo code on how the model runs; this is followed by Section 4, which contains the code and experimentation description; next, in Section 5, the results are shown in detail and a discussion takes place; and finally, Section 6 will close with the general conclusions of the model and some recommendations for future work.

## 2. Related Work

Even though LDA is a good topic model, it has been proven that applying modifications to the base model can provide additional functionality and improve its performance, as shown in this section.

One of the more interesting modifications of the canonical LDA models is based on the dynamic versions, where a time series is assumed for each topic, thus resulting in the assumption that the topic might have variations over time. The Dynamic Latent Dirichlet

Allocation (DLDA) was proposed by Blei, et al. [20] to make LDA have variations on their topics based on time. In terms of conversational text, a modification to DLDA is Conceptual Dynamic Latent Dirichlet Allocation (CDLDA) [21], which detects topics using a bag of words and then implementing temporal features. However, since DLDA and CDLDA are LDA-based models, they suffer from the same core issues as LDA that were mentioned in the previous section. Using the same intuition from the DLDA and having assured that the ETM currently outperforms LDA, a modification for the ETM (with a dynamic perspective) was implemented [22].

Looking at the solutions from another perspective, the Locally Consistent LDA (LCLDA) was an approach designed to improve the conventional LDA model by using an embedding layer with K-means based on the bag of words and improving the solving method [23]. Other models that also used word vector spaces proved to be great additions for the current modifications [24,25]. There have also been some advances regarding the BERT architecture [26] for the embeddings and using this embedding layer to create topic models, such as the Combined Topic Model (CTM) [27] and Zero-shot Cross-lingual Topic Model (ZeroShot-CTM) [28]. All these models, although improving on LDA, still lack the possibility of using the order of the words to arrive at better predictions.

In the area of Neural Topic Modelers (NTMs), which refers to the neural network models used for topic modelling, there is a model for Sentence Generating Neural Variational Topic Model (SenGen), which uses RNN for creating the vector of topics based on sampling sentences instead of a bag of words [29]. Continuing the trend with LSTM, there are topic models that approach the problem by using LSTMs to understand the word sequences of the topic and dynamically model them [30,31], until finally arriving at the more advanced LSTM topic model with attention mechanism (Topic Attention Networks for Neural Topic Modelling), which explores the combination of both approaches in the topic–document distribution [32]. Although NTMs work great, word embeddings are not used to improve the input of the models over raw text.

## 3. Fundamentals

The Recurrent Embedded Topic Model (RETM) is based on the current ETM (which is based on a combination of LDA and CBOW) and Recurrent Neural Networks (i.e., LSTM). Providing some context on the models, we assume that we have a corpus of $M$ documents $D = \{d_1, d_2, \ldots, d_M\}$, we also assume there is a vocabulary which is indexed from $\{1, 2, \ldots, V\}$. A document is a sequence of $n$ words denoted by $w \in \{w_1, w_2, \ldots, w_n\}$ where $w_n$ is the nth word in the sequence. We also assume that $K$ topics are provided as a hyperparameter to the model ($k \in \{1, 2, \ldots, K\}$) and associated with a learnable distribution over the vocabulary ($\beta_k$ for each topic $K$).

The following Sections 3.1–3.5 will provide an overview of the models that the RETM is based on, providing a mathematical description of the rationale behind the model and the history of the previous models with the modifications that have been undertaken that lead to the RETM.

### 3.1. Latent Dirichlet Allocation

LDA is a generative probabilistic model of documents and it assumes the following process for each document $d$ in the corpus $D$ [5]:

1. Choose $\theta \sim \text{Dir}(\alpha)$.
2. For each topic $k$, $\gamma \sim \text{Dir}(\beta)$.
3. For each word $w$:

   (a) Choose a topic $k \sim \text{Multinomial}(\theta)$.
   (b) Choose a word $w \sim \text{Multinomial}(\gamma_k)$.

For LDA, $\alpha$ and $\beta$ are the hyperparameters that Dirichlet needs in order to properly work.

### 3.2. Continuous Bag of Words

CBOW is a neural network that trains a word $w_z$ based on the context of which words came after and before that word ($w_{z-1}$, $w_{z-2}$, $w_{z+1}$, $w_{z+2}$, etc.). These contextual words are passed through a dense Neural Network (NN) that aims to predict the word $w_z$. If we look at it this way, we can notice that the vector (neurons) from the embedding layer are the ones representing the word $w_z$, where, using those neurons, you can represent the vocabulary; please refer to Figure 2 for visual aid. Given a word $w_z$ and calculating the cosine distance [33] in the embedding layer for each word, they were able to find similar words to $w_z$ and that, precisely, is the breakthrough in CBOW [34]. Regarding the equation, the embedding matrix $\rho$ is a $L \times V$ matrix where $\rho_w \in \mathbb{R}^L$ (it is important to note that $L$ refers to the number of neurons on the embedding layer). The context embedding is the sum of the embedding vectors ($w_{n-1} \dots$ for the words surrounding each word $w_n$) and is represented by $\alpha_w$. Inside the current model, the CBOW is defined as (1):

$$w \sim \text{softmax}\left(\rho^{\top} \alpha_w\right) \tag{1}$$

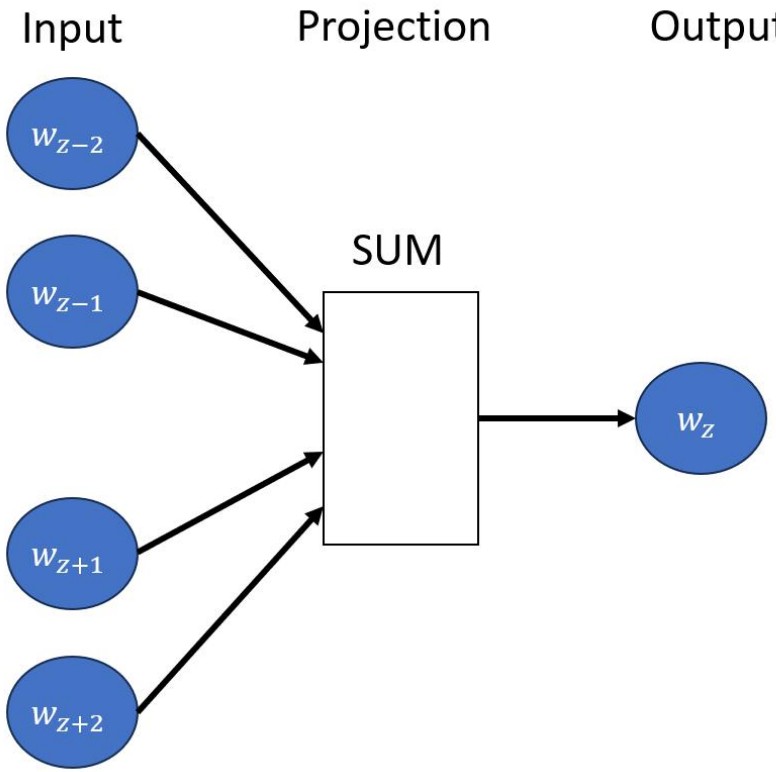

**Figure 2.** CBOW model. Adapted from [14].

### 3.3. Embedded Topic Modelling

The ETM [17] is a model that uses an embedding layer for the vocabulary while the distribution of the $K$ topics remain similar to the LDA model, however the $k$th topic is a vector $\alpha_k \in \mathbb{R}^L$ in the embedding space where $\alpha_k$ is a distributed representation in the semantic space of words. The ETM, being a log linear model, in its generative process uses the inner product of word embeddings and the topic embeddings, assigning high probabilities to the words $w$ in topic $k$ by measuring the agreement between the word's embedding and the topic's embedding.

Similar to the LDA, for each of the $d$ documents in the corpus $D$:

1. Choose $\theta \sim \mathcal{LN}(0, I)$.
2. For each word $w$:
   (a) Choose a topic $k \sim \text{Multinomial}(\theta)$.
   (b) Choose a word $w \sim \text{softmax}(\rho^\top \alpha_k)$.

The modification in $\theta_d$ is changing the Dirichlet process to a logistic normal distribution and it was carried out to easily reparametrize the inference algorithm (based on variational inference) [17]. If analyzed closely against LDA and CBOW, you can easily spot the changes on the distribution Dirichlet for $\theta_d$ and the substitution of the word distribution for the softmax of the embedded layer.

### 3.4. LSTM

Speaking of recurrent neural networks, since the recurrency is for the $L$ dimension, the exploding/vanishing problem easily arrives at the RETM. Given this, the model uses the LSTM neural networks for the recurrency in the calculation of $\theta_d$. The LSTM model was initially proposed by Hochreiter et al. [35] and the following definition of the model will be based on Graves' book [36]. For ease of the explanation, we will just go through the forward pass, stating that we recommend going through the backward pass involved during the training process from the original paper. We call $m_{ij}$ as the weight of the connection between unit $I$ and unit $J$, we also denote $s_c^t$ as the state of cell $c$ (this refers to one of the $C$ memory cells) at time $t$, $b_j^t$ as the network input for unit $j$ in time $t$ and $a_j^t$ as the activation for unit $j$ in time $t$. $\iota$, $\phi$ and $\omega$ refer to the input gate, forget gate and output gate of the block. The peephole weights (previous internal states and hidden states) from cell $c$ for the input, forget and output gates are denoted as $m_{c\iota}$, $m_{c\phi}$ and $m_{c\omega}$ respectively. In the context of the current work, $x^t = \rho^\top \alpha_{w_t}$. Finally, $f$ is the activation function of the gates, $g$ and $h$ are the cell input and output activation functions. To simplify, a single cell of the LSTM can be observed in Figure 3.

Input gates

$$a_\iota^t = \sum_{i=1}^{I} m_{i\iota} x_i^t + \sum_{h=1}^{H} m_{h\iota} b_h^{t-1} + \sum_{c=1}^{C} m_{c\iota} s_c^{t-1}$$
$$b_\iota^t = f\left(a_\iota^t\right)$$

Forget gates

$$a_\phi^t = \sum_{i=1}^{I} m_{i\phi} x_i^t + \sum_{h=1}^{H} m_{h\phi} b_h^{t-1} + \sum_{c=1}^{C} m_{c\phi} s_c^{t-1}$$
$$b_\phi^t = f\left(a_\phi^t\right)$$

Cells

$$a_c^t = \sum_{i=1}^{I} m_{ic} x_i^t + \sum_{h=1}^{H} m_{hc} b_h^{t-1}$$
$$s_c^t = b_\phi^t s_c^{t-1} + b_\iota^t g\left(a_c^t\right)$$

Output gates

$$a_\omega^t = \sum_{i=1}^{I} m_{i\omega} x_i^t + \sum_{h=1}^{H} m_{h\omega} b_h^{t-1} + \sum_{c=1}^{C} m_{c\omega} s_c^t$$
$$b_\omega^t = f\left(a_\omega^t\right)$$

Cell outputs

$$b_c^t = b_\omega^t h\left(s_c^t\right)$$

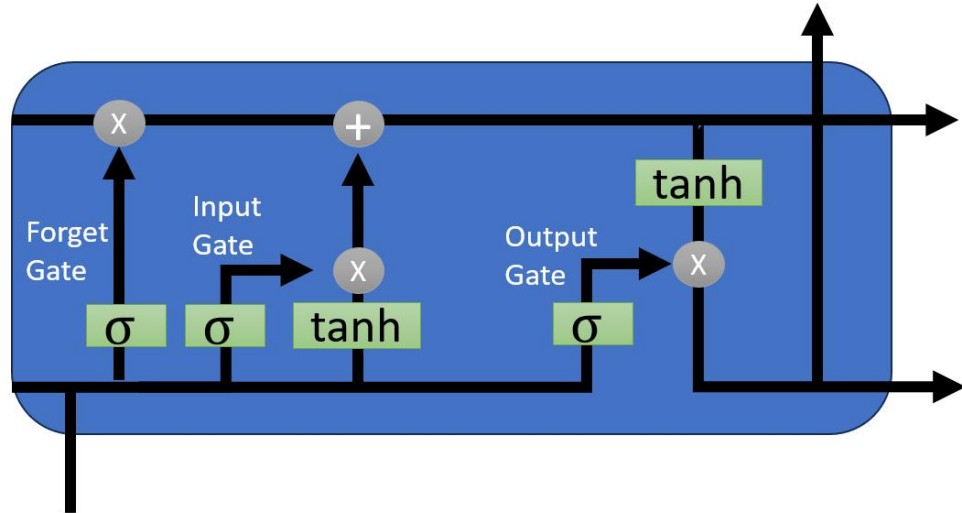

**Figure 3.** LSTM model, single cell. Adapted from [36].

### 3.5. Inference and Estimation

In Equation (2), the parameters $\rho$ and $\alpha$ determine the word distribution; furthermore, $\delta_d$ is a Gaussian seed during the generative process. The inferential problem to be solved in this model is calculating the posterior distribution of the variables for any given document, although this distribution is computationally unrealistic to achieve (due to the integral, as we are sampling a continuous variable), such as is shown in (2) [17].

$$p(\mathbf{w}_d \mid \alpha, \rho) = \int p(\delta_d) \prod_{n=1}^{N_d} p(w_{dn} \mid \delta_d, \alpha, \rho) \mathrm{d}\delta_d \tag{2}$$

Like LDA, ETM uses variational inference to get around calculating the previous equation and going through proven approximations [37]. They used amortized inference where the variational distribution of $\delta_d$ depends on $w_d$ and the variational parameters $v$, where a Gaussian distribution whose mean and variance comes from a neural network parametrized by $v$ (when $w_d$ is ingested it outputs the mean and variance of $\delta_d$). The ELBO (Evidence Lower Bound) is the function of the model parameters and variational parameters that is optimized throughout the training of the model, as shown in (3) [17].

$$\begin{aligned}
\mathcal{L}(\alpha, \rho, v) = &\sum_{d=1}^{D} \sum_{n=1}^{N_d} \mathbb{E}_q[\log p(w_{nd} \mid \delta_d, \rho, \alpha)] \\
&- \sum_{d=1}^{D} \mathrm{KL}(q(\delta_d; w_d, v) \| p(\delta_d)).
\end{aligned} \tag{3}$$

This equation helps $\delta_d$ to place mass on topic proportions to explain the observed words where the Kullback Leibler (KL) divergence helps them get closer to the prior $p(\delta_d)$. This metric supports the maximization of the expected complete log-likelihood $\sum_d \log p(\delta_d, w_d \mid \alpha, \rho)$ [17].

The next Algorithm 1 assumes that NN$(x; v)$ refers to a neural network (linear) where $x$ is the input and $v$ are the variational parameters of the network:

---

**Algorithm 1** ETM

---

1: Initialize model and variational parameters
2: **for** epoch **do**
3:     Compute $\beta_k = \text{softmax}(\rho^\top \alpha_k)$ for each topic $k$
4:     Choose a minibatch **d** of documents
5:     **for** each document $d$ in **d do**
6:       Get normalized bag-of-word. $\mathbf{x}_d$
7:       Compute $\mu_d = \text{NN}(\mathbf{x}_d; \nu_\mu)$
8:       Compute $\Sigma_d = \text{NN}(\mathbf{x}_d; \nu_\Sigma)$
9:       Sample $\theta_d \sim \mathcal{LN}(\mu_d, \Sigma_d)$
10:       **for** each word $w$ in the document $d$ **do**
11:         Compute $p(w \mid \theta_d) = \theta_d^\top \beta \cdot, w$
12:       **end for**
13:     **end for**
14:     Estimate the ELBO and its gradient (backprop.)
15:     Update model parameters $\alpha_{1:K}$
16:     Update variational parameters $(\nu_\mu, \nu_\Sigma)$
17: **end for**

---

*3.6. Recurrent Embedded Topic Modelling*

In this paper, the Recurrent Embedded Topic Model (RETM) is proposed to enhance the quality of the topics for data analysis and data organization purposes by reusing the embedding layer into a recurrent neural network (LSTM). The rationale behind is based on trying to extract the meaning of the word currently analyzed by each of the values it has on the $L$ dimensional space of the CBOW. To simplify, since it has been proven that the CBOW model contains the "meaning" of the word in an $L$ dimensional space [34], then using the recurrency of the order of the words in the text can help the model to provide "context" when determining the topic distributions. Thus, given the recurrency of the $L$ dimensional space inside an LSTM, the neural network can further predict $\theta_d$, which corresponds to the topic–document distribution of the model. As it can be seen in the pseudo code 2 after obtaining the normalized bag of words for each document on the minibatch, the embedded layer is used to generate the third dimension of the tensor (the recurrency layer of the LSTM). This means that the shape of the tensor is $[w_{dn}, \mathbf{d}, L]$, and it is important to note that **d** is defined in the pseudo code as the mini batch of documents and $w_n$ is referring to maximum number of words $w$ that exist in $D$. After the tensor is created, the LSTM is computed with the variational parameters and then a dropout is executed to prevent overfitting. Next, the computations of $\mu_d$ and $\Sigma_d$ are calculated to sample using a logistic normal distribution, $\theta_d$. Prior to the calculation of the ELBO, the tensor must be returned to a two-dimensional space to be able to multiply against $\beta$, this reduction of the dimensionality is made with an aggregation based on max (the aggregation is further explain after Algorithm 2), until finally arriving at the calculation of the ELBO and the gradient.

Coming back to the training and inference/estimation of the model, it remains the same as what the ETM has already proposed.

Algorithm 2 shows how the model works step by step. Some recommendations to ensure that the RETM works properly are to use only a text column (discard everything else) and apply some cleaning techniques to the raw text prior to using the RETM.

After the LSTM calculations (in step number 15), where $\theta_d$ is a third-dimensional tensor) and prior to computing the ELBO, $\theta_d$ must be transformed to a two-dimensional tensor. To achieve this two-dimensional tensor, an aggregation was applied by using max. Max was used to prioritize the most salient words for each topic creating clear cut topics. It is important to know that this strategy can create bias to extreme values. If using another aggregation, min can create more general topics with the downfall of hitting the topic coherence metrics where the words will be more dispersed between them. Average can

provide a balanced representation of the topics leading to more comprehensive topics. In the end, max was used since it provided better perplexity scores, but the door is not closed for more experimentation on the reduction of the recurrent dimension.

---

**Algorithm 2** RETM

---

1: Generate train and test datasets plus the vocabulary vector
2: Calculate the CBOW or use the standard one
3: Initialize model and variational parameters
4: **for** epoch **do**
5:     Compute $\beta_k = \mathrm{softmax}(\rho^\top \alpha_k)$ for each topic $k$
6:     Choose a minibatch $\mathbf{d}$ of documents
7:     **for** each document $d$ in $\mathbf{d}$ **do**
8:         Get normalized bag-of-word. $\mathbf{x}_d$
9:         Add the extra dimension to $\mathbf{x}_d$ by ordering the words and extracting their values from the CBOW
10:         Compute $\zeta_d = \mathrm{LSTM}(\mathbf{x}_d; \nu_\zeta)$
11:         Create dropout $\nu_\zeta$
12:         Compute $\mu_d = \mathrm{NN}(\mathbf{x}_d; \nu_\mu)$
13:         Compute $\Sigma_d = \mathrm{NN}(\mathbf{x}_d; \nu_\Sigma)$
14:         Sample $\theta_d \sim \mathcal{LN}(\mu_d, \Sigma_d)$
15:         Return matrix to 2 dimensional by applying aggregation (max) for $\theta_d$
16:         **for** each word $w$ in the document $d$ **do**
17:             Compute $p(w \mid \theta_d) = \theta_d^\top \beta \cdot, w$
18:         **end for**
19:     **end for**
20:     Estimate the ELBO and its gradient (backprop.)
21:     Update model parameters $\alpha_{1:K}$
22:     Update variational parameters $(\nu_\zeta, \nu_\mu, \nu_\Sigma)$
23: **end for**

---

## 4. Code and Experimentation

As previously stated, the current model and code act as a modification to the current ETM model that was originally proposed by Dieng et al. in their paper Topic Modelling in Embedding spaces [17]. This paper also includes an original version of the code that can be found on Github [38], but although all the core code is inside the Github, in this paper we preferred to use a modification of the code that was created by Luis Mateos, which provided some adaptations and an overall better structure [39]. To be specific, some of the changes inside the code for this paper were based on the architecture of the model. The perplexity calculation inside the training dataset was captured by reusing the existing calculations' (Reconstruction loss) that were happening on each epoch. Another of the big changes has to do with the creation of a new pipeline for loading and transforming (capitalization, stopwords removal and punctuation/special characters removal) the datasets. After the data are ready to be used, the models were fitted, evaluated and compared to each other (LDA, ETM, Zeroshot-CTM, CTM and RETM). The data pipeline, model fitting and evaluation can be found in main.py inside the repository. Requirements.txt lives inside the repository for help on easy installation. Github is available for this code [40]. Finally, all the code was implemented on Pytorch, where the stochastic optimization was used with Adam [41] for the learning rate and using the ELBO as the metric to be optimized on the NN (RETM).

Among the models that are used for comparison to RETM, we can find CTM and Zeroshot-CTM, which were both taken from Github [42]. Although there is no direct implementation of the perplexity metrics for CTM and Zeroshot-CTM, the perplexity metric was calculated by reusing the reconstruction loss, in the same way that the ETM

and RETM calculate theirs. In the scenario of LDA, the model was used from the Github page of ScikitLearn [43].

For the datasets used in this paper, we compare the performance of the five models in different applications of what unstructured text data might look like. The way we compared the performance was by fitting and predicting the models on different datasets and extracting their perplexity and topic coherence metrics (same data and same process for all five models). The datasets range between emails [44], genre classification of movies from IMDb [45], book summaries [46] and news [47]. All these datasets vary drastically between each other, from the words and sentence structure to the number of words in each document (rows) and the probability of occurrence of each of the words inside the text, as can be seen in Table 2. Although the datasets provide thousands of rows to use, given computational limitations the amount rows that were used was lowered to 30,000, where in experimental fashion, there was small differences comparing to full dataset for each file were used (the shuffle was conducted with random seed to ensure repeatability).

**Table 2.** Table shows descriptive data of the datasets used on the experiments.

| Dataset | Total Number of Words | Average Number of Words |
|---|---|---|
| Movies | 170,085 | 58 |
| Email | 40,557 | 166 |
| Books | 117,764 | 234 |
| News | 79,150 | 4066 |

The computer that was used to create and run all models has the following specs: System Manufacturer = Gigabyte, System Model = Aorus 15P XC, BIOS = FB07, Operating System = Windows 11 Home, Python 3.9 (plus all the dependencies that can be found on the requirements.txt of the repository), RAM = 32 GB, CPU = Intel Core i7-10870H and GPU = Nvidia GeForce RTX 3070 Laptop.

After the dataset is loaded into memory, a small preparation of the text was conducted prior to using it on the model. The text transformation pipeline was as follows:

1. Apply lower capitalization.
2. Remove stop words using nltk.corpus.
3. Remove all numbers and punctations (including special characters).
4. Randomize all rows with a random seed = 42.
5. Extract the top 30,000 rows.

The metric that was used to compare the performance of the model was the perplexity, which represents one of the most important metrics for topic modelling. Perplexity can be defined as the exponentiated average negative log-likelihood of a sequence [48], as shown in (4). Another metric that was used to analyze the topics that were created by the Zeroshot-CTM, CTM, ETM and RETM was the topic coherence. Topic coherence is a measure to understand how close the words inside a topic (the closer the words are, the more condensed the topic is). It is calculated by the Pointwise Mutual Information [49], which is used to measure the connection between two things (in this case *x* and *y*), as shown in (5) (this equation was taken from Wu et al. [50]).

$$\text{perplexity}\left(\boldsymbol{n}^{\text{test}}, \boldsymbol{\lambda}, \alpha\right) \triangleq \exp\left\{-\left(\sum_i \log p(n_i^{\text{test}} \mid \alpha, \boldsymbol{\beta})\right) \Big/ \left(\sum_{i,w} n_{iw}^{\text{test}}\right)\right\} \tag{4}$$

$$\text{PMI}(x, y) = \log \frac{p(x, y)}{p(x)p(y)} = \log \frac{p(x \mid y)}{p(x)} = \log \frac{p(y \mid x)}{p(y)} \tag{5}$$

## 5. Results

In this section, the RETM is compared based on perplexity and topic coherence against other models using the datasets that were previously exposed in Section 4. Table 3 demon-

strates how the RETM stacks against the other models (CTM = more than 5162 times better, Zeroshot-CTM = more than 5336 times better, LDA = more than 25 times better and ETM = more than 15 times better) using perplexity (where lower is better) on the training dataset by proving to be extremely efficient at creating quality topics thanks to its recurrency on the neural network calculation.

**Table 3.** Table shows the perplexity values on training datasets used on the experiments.

| Dataset | LDA | CTM | Zeroshot-CTM | ETM | RETM |
|---------|-----|-----|--------------|-----|------|
| Movies | 4557.30 | 473,741.98 | 490,147.51 | 632.4 | 39.6 |
| Email | 2612.94 | 301,272.40 | 477,580.90 | 763.2 | 34.5 |
| Books | 4524.45 | 243,154.4 | 315,798.9 | 1675.5 | 47.1 |
| News | 2400.48 | 607,338.92 | 495,231.35 | 4191.4 | 92.8 |

On the test dataset, the RETM still proves to outperform CTM (more than 121 times better), Zeroshot-CTM (more than 71 times better), LDA (more than 1689 times better) and ETM (between 10% and 90% better), as shown by Table 4.

**Table 4.** Table shows the perplexity values on test datasets used on the experiments.

| Dataset | LDA | CTM | Zeroshot-CTM | ETM | RETM |
|---------|-----|-----|--------------|-----|------|
| Movies | $8928.18 \times 10^5$ | 463,774.15 | 476,982.15 | 647.2 | 328.3 |
| Email | 5,964,502.4 | 245,077.4 | 407,738.1 | 758 | 516.8 |
| Books | 2,068,412.6 | 203,167.4 | 333,945.1 | 1717.6 | 1224.5 |
| News | $8580.2 \times 10^{13}$ | 591,459.6 | 349,622.3 | 5396.5 | 4878.8 |

Speaking of topic coherence (which refers to how condensed the words are between others), there is small difference between the RETM and the ETM in the books, news and movie datasets, but a significant difference for the email dataset, as shown by the Figure 4. Zeroshot-CTM and CTM are far away from being good models compared to ETM and RETM. As a side note, for topic coherence, more is better.

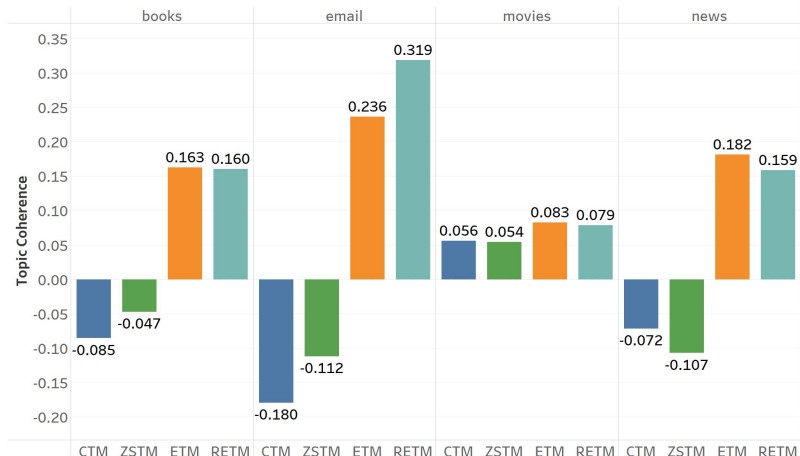

**Figure 4.** Comparative between CTM, ZeroShot-CTM, ETM and RETM using the topic coherence metric over different datasets (emails, book summaries, news and genre classification of movies).

Finally, Figure 5 shows a single topic, using the RETM model, from the book's dataset. T-distributed Stochastic Neighbour Embedding (TSNE) [51] was implemented over the *L* dimensional vector of the embedding to reduce the dimensionality to two dimensions and create a visualization of how the words look for the topic (and the distances between them). It shows the top 20 words for the topic, where each word has a different color and size (where bigger the size, the more impact it has over the topic distribution).

As shown in the graph, this topic must talk about books with stories from the medieval era (probably) and as expected, most words are near each other.

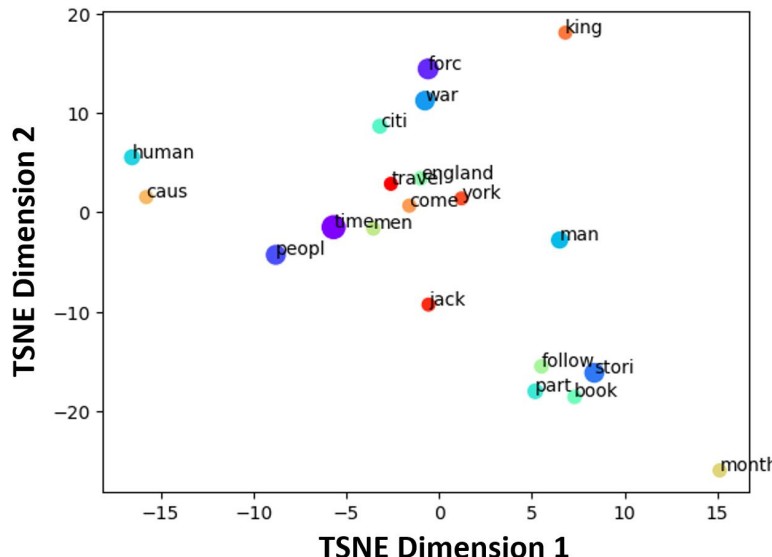

**Figure 5.** Two-dimensional representation of the embedding space where the top 20 words for a topic are shown. Each word is represented by a color and the size of the circle corresponds to the importance of that word at describing the topic.

*Discussion*

Pairing the results of the train data set, it showed an improvement of at least 15 times better perplexity values over CTM, Zeroshot-CTM, ETM and LDA, meaning that every time a text-based dataset is needed for data analysis, data organization or any task where no prediction is needed over new data, this model proved to heavily outperform current models. On the other hand, the perplexity calculated in test dataset, although, still better than all models, proved to not be as high as the training dataset, being at least 10% better than the other models.

In terms of topic coherence, the RETM showed slightly worse results for the books (RETM = 0.1602, ETM = 0.1630), movies (RETM = 0.0789, ETM = 0.0828) and news (RETM = 0.1587, ETM = 0.1816) datasets, but better for the email (RETM = 0.3188, ETM = 0.2364) dataset; however, it still severely outperformed CTM and Zeroshot-CTM, showing a possible hypothesis that since the email dataset has a much lower vocabulary, this could have affected the predictions for the RETM, since $\rho$ has less coordinates in $V$ which could increase the convergence in $\alpha_w$. Unfortunately, there is not sufficient data to prove this hypothesis, and so it will be highly encouraged as future work.

Outside of previously stated results, it is important to mention that one of the biggest limitations to the RETM is represented by the high computational cost it needs to fit to a dataset. As the recurrence of the model is based on the order of the words per text, coupling this with the embedding layer, if using the 300-dimensional space (as it was the case for these experiments), the amount of V-RAM that the GPU (Nvidia GeForce RTX 3070 Laptop) needs to process the model represents a high cost. To reduce the cost of V-RAM, the batch size can be lowered, but it can penalize the final scores. Once the batch sizes are lowered, it is normal to expect a higher time to compute the model, although it is faster than the traditional LDA calculations on big vocabularies. Also, to reduce computational power, the average amount of words in the dataset was used as the size of the vector for the topic–document distribution calculation, and the door is still open for running the model using the full size vector, given the hypothesis that this can also prove beneficial for the results of the model. Running the current model on a CPU seemed to be a never-ending task. To provide an example, the movie dataset (which contains 30,000 rows worth of text

data), using a batch size of six, with eight neurons inside the LSTM, cost 5 GB of RAM and 4 GB of VRAM from the GPU and took 28 min to fit the model; in comparison, the ETM, for the same dataset, took 1.5 GB of RAM and 2.3 GB of VRAM and only spent 1 min fitting the model.

LSTM has opened the door for several applications in the topic modelling area. Given that the aggregation level for the LSTM on this model was based on max this created clear-cut topics and improved the quality in almost all datasets; however, there is still room for experimentation and different uses cases for the current RETM (different aggregations) that can benefit from the current implementation, making a less static model compared to the rest.

The RETM was shown to be outstanding at topic modelling tasks: any database that uses text-based data can benefit from the implementation of this model, although it may be penalized with small vocabularies. Since the RETM can calculate its own CBOW, it can provide useful topic modelling in domain-specific documents similar to the ETM. In the end, the RETM heavily outperformed previous methodologies and proved to be a state-of-the-art model for topic-modelling problems.

## 6. Conclusions

The Recurrent Embedded Topic Model was proposed in this paper with the aim of using the order of the words in an LSTM to provide more accurate representations of the topics. Four experiments were conducted using text-based datasets (email data, movie genre classification, news and book summaries) with different distributions and vocabularies, to test the model against CTM, Zeroshot-CTM, LDA and ETM. The metrics that were used to test the model were perplexity and topic coherence and the model proved to outperform all models in almost all scenarios. Although there were great results from the RETM, the full power of the model still remains in the shadows (based on the computational limitations that were faced), however the RETM is a state-of-the-art model for conducting topic proportion distributions and is highly recommended to extract quality-based topics out of the documents.

The Recurrent Embedded Topic Model has proved to be a computationally expensive model due to the recurrence of the word order applied on the neural network architecture. As of the writing of this paper, the batch size remained consistently low due to the amount of V-RAM required for the model, and the average of the words inside the texts were used to create the three-dimensional vector for the LSTM, so the hypothesis of how the model would work on bigger GPUs remains unanswered.

Recalling the embedded layer, there was not any detailed work on the standard natural language processing pipeline that was applied to the text prior to introducing the data into the model. There could be some potential improvement while calculating the CBOW over cleaned text, providing another hypothesis on how to improve the current model.

**Author Contributions:** Conceptualization, C.V.; methodology, H.P.; software, C.V.; validation, C.V.; formal analysis, C.V.; investigation, C.V.; resources, C.V.; data curation, C.V.; writing—original draft preparation, C.V. and H.P.; writing—review and editing, H.P.; visualization, C.V.; supervision, H.P.; project administration, C.V. All authors have read and agreed to the published version of the manuscript.

**Funding:** This research received no external funding.

**Informed Consent Statement:** Not applicable.

**Data Availability Statement:** All the code and data needed to replicate the results can be found at: Vargas, Carlos. Recurrent Embedded Topic Model. *Github* **Published on 9 August 2023**, Available online: https://github.com/NxrFesdac/RETM (accessed on 13 October 2023). The data was extracted from the following sources: Saber_Cali; kagglethebest. Spam email data original and CSV file (Spamassassin). *Kaggle* **2023**, Available online: https://www.kaggle.com/search?q=Spam+Email+Data+original+%26+CSV+file+(Spamassassin) (Downloaded on 23 June 2023). Radmirkaz. Genre Classification Dataset IMDb. *Kaggle* **2021**. Available online: https://www.kaggle.com/datasets/hijest/genre-classification-dataset-imdb (Downloaded on 23 June 2023). Bamman, David; Jasminyas.

CMU Book Summary Dataset. *Kaggle* **2018**. Available online: https://www.kaggle.com/datasets/ymaricar/cmu-book-summary-dataset (Downloaded on 23 June 2023).

**Acknowledgments:** We would like to express our sincere appreciation to Colegio de Matemáticas Bourbaki, where Alfonso Ruiz imparts his unparalleled knowledge, whose unwavering guidance and exceptional teaching in mathematics have been instrumental throughout this journey. The academic environment that Colegio de Matemáticas Bourbaki has provided has been the perfect incubation for intellectual growth and exploration.

**Conflicts of Interest:** The authors declare no conflict of interest.

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
