# Peer review of "Recurrent Embedded Topic Model"

_applsci, doi:10.3390/app132011561_

Round 1

Reviewer 1 Report (New Reviewer)

The authors propose an extension to the existing algorithms for topic modeling, using LSTM-based recurrent neural networks. The proposed method shows significant improvements in several test cases.

The article is clearly written, deeply analyzes existing algorithms and explains the proposed method in great detail. Experimental results are clearly described. Some suggestions for further improvement follow:

1) "zetabytes" at line 18 and in caption of Figure 1: the correct spelling is "zettabytes".

2) Section 3.4: even if LSTM is a well-known element, I suggest adding a figure with a simplified depiction of the blocks representing the formulas, to have a better overall understanding of its architecture.

3) Figure 2 and 3 appear quite blurry. Since they are copied from other articles, please capture them with a larger resolution or, even better, in vector format.

4) Figure 1, 2, 3: you should perhaps state, in the caption, the copyright and license of the pictures (this can be better confirmed by the Editor).

5) Figure 4: labels are hardly readable, even when zooming in. Since it's supposedly generated by you and thus easily modifiable, I suggest recreating it with clearer text. Moreover the picture would be clearer with a label on y axis, like for example "topic coherence" (or what it's more appropriate).

6) Similarly, in Figure 5 it's not very clear what numbers on x and y axes represent. I suggest adding labels.

7) The sentence at lines 52-56 is not very clear, can you better rephrase it?

8) The sentence at lines 185-187 ("Recalling the LSTM model...") seems missing its continuation. It should be rewritten with a different verb tense or tied to adjacent sentences.

9) Line 177: $\theta_d$ has not been clearly defined in previous formulas (what's the "d" subscript?)

10) Equation 2 contains several quantities not clearly defined at that point and it's thus confusing. Please explain if it's somehow linked to the previous sections and/or add a brief explanation of the involved variables. This would greatly help the understanding of the following equations and algorithms, too.

11) At line 222 you say "the embedded layer is used to generate the third dimension of the tensor". Since setting the right matrix shape is a significant problem in the NN field, can you elaborate this step a little more with the rationale (much like you do for the opposite step)? Also (optionally) it could be useful to indicate the resulting shape of matrices in Algorithm 2, perhaps as comments.

12) At line 340 you discuss hardware costs and limitations, and the needed trade-offs. Since this might be interesting for people wanting to reuse your code, can you add specific information? For example: memory used, computation time, etc., also for several choices of parameters (batch size, ...), together with their effect on accuracy.

Some sentences need to be rephrased. Some overall minor editing is needed.

Author Response

COVER LETTER

Oct 17th, 2023

Applied Sciences

Dear Reviewer,

We are pleased to submit our revised manuscript entitled “Recurrent Embedded Topic Model” for possible inclusion in the journal Applied Sciences.

In this revised version, we improved the manuscript considering the reviewers’ comments. Thank you for the valuable feedback.

Below, you will find the point-to-point revision of the comments.

Reviewer’s comments

Answers to Reviewers

The authors propose an extension to the existing algorithms for topic modeling, using LSTM-based recurrent neural networks. The proposed method shows significant improvements in several test cases.

The article is clearly written, deeply analyzes existing algorithms and explains the proposed method in great detail. Experimental results are clearly described. Some suggestions for further improvement follow:

Thank you for your valuable comments, everything you said here is correct and we appreciate all the hard work that was placed into your review.

"zetabytes" at line 18 and in caption of Figure 1: the correct spelling is "zettabytes".

Changes applied

Section 3.4: even if LSTM is a well-known element, I suggest adding a figure with a simplified depiction of the blocks representing the formulas, to have a better overall understanding of its architecture.

A new figure was created to show the LSTM cell with reference to the gates that are described in the equations of the paper.

 Figure 2 and 3 appear quite blurry. Since they are copied from other articles, please capture them with a larger resolution or, even better, in vector format.

Figure 2 was fixed and now is fully visible.

Figure 3 was remade from scratch and adapted to the equation variables that are defined in the paper.

Figure 1, 2, 3: you should perhaps state, in the caption, the copyright and license of the pictures (this can be better confirmed by the Editor).

Figure 1 was adapted into a table and provided reference to the original work.

Figure 2 is an open-access article which permits unrestricted use, distribution, and reproduction in any medium as long as the original work is properly cited (which is our current scenario). (https://direct.mit.edu/tacl/article/doi/10.1162/tacl_a_00325/96463/Topic-Modeling-in-Embedding-Spaces )( https://creativecommons.org/licenses/by/4.0/legalcode )

Figure 3 was adapted to an image with the equations that are defined in the paper, also there is reference to the original work.

Editor also asked for the copyright, we are answering the editor at the beginning of this letter. Thank you for pointing this out!

Figure 4: labels are hardly readable, even when zooming in. Since it's supposedly generated by you and thus easily modifiable, I suggest recreating it with clearer text. Moreover, the picture would be clearer with a label on y axis, like for example "topic coherence" (or what it's more appropriate).

The new Figure (we increased all text sizes, and the text is now bold) should be clearer, also y axis label was added.

Similarly, in Figure 5 it's not very clear what numbers on x and y axes represent. I suggest adding labels.

Figure 5 is using the TSNE technique for dimensionality reduction. Since the embedding space is a 300-dimensional vector, TSNE = 2 was used to show a visual representation of the space for a single topic. Labels were added to x and y axis, although, there is nothing much to add except than “TSNE Dimension 1” and “TSNE Dimension 2”

The sentence at lines 52-56 is not very clear, can you better rephrase it?

It was rephrased, changes can be visualized in blue

The sentence at lines 185-187 ("Recalling the LSTM model...") seems missing its continuation. It should be rewritten with a different verb tense or tied to adjacent sentences.

It was rephrased, changes can be visualized in blue

Line 177: $\theta_d$ has not been clearly defined in previous formulas (what's the "d" subscript?)

d corresponds to a single document in corpus D. Where fundamentals start it is explained D and d, also d is shown at the beginning of the ETM equation as:

“Similar to the LDA, for each of the d documents in the corpus D:

1. Choose θ ∼ LN (0, I)…..”

Equation 2 contains several quantities not clearly defined at that point and it's thus confusing. Please explain if it's somehow linked to the previous sections and/or add a brief explanation of the involved variables. This would greatly help the understanding of the following equations and algorithms, too.

Thank you for pointing it out, it was rephrased to define all the variables that were not previously defined. These changes can be seen in blue.

At line 222 you say, "the embedded layer is used to generate the third dimension of the tensor". Since setting the right matrix shape is a significant problem in the NN field, can you elaborate this step a little more with the rationale (much like you do for the opposite step)? Also (optionally) it could be useful to indicate the resulting shape of matrices in Algorithm 2, perhaps as comments.

The rationale was further expanded, thank you for the comment.

I agree with the importance of explaining the matrix shape for a NN. A new line explaining what the shape of the matrix, was added in blue to explain what is the shape needed in the LSTM.

At line 340 you discuss hardware costs and limitations, and the needed trade-offs. Since this might be interesting for people wanting to reuse your code, can you add specific information? For example: memory used, computation time, etc., also for several choices of parameters (batch size, ...), together with their effect on accuracy.

Some comments have been added to further explain the computational difficulties that the model has. This can be seen in blue.

All the changes were highlighted in blue text color in the document file.

We are looking forward to hearing from you soon.

Best regards,

Carlos Vargas Fraga

Reviewer 2 Report (Previous Reviewer 1)

The main concerns pointed out in the last round of review by this reviewer centred on:

1. the performance improvement of RETM wrt ETM, given the added complexity (Figure 5 in previous version)

2. the worse coherence of RETM vs existing models (Figure 6 in previous version).

The authors provide detailed comments or clarifications on these, instead change the presentation of results from in said figures to tables (cf. Tables 2 & 3). Moreover, the presentation of relative performance can be confusing to readers! Is higher or lower perplexity/coherence scores preferred? Review lines 303 - 310: saying method X = more than Y times better, etc. Is this consistent with direction of score preference?

Acceptable in general. Please avoid the phrase "moving on to ..." that is informal/oral presentation style.

Author Response

COVER LETTER

Oct 17th, 2023

Applied Sciences

Dear Reviewer,

We are pleased to submit our revised manuscript entitled “Recurrent Embedded Topic Model” for possible inclusion in the journal Applied Sciences.

In this revised version, we improved the manuscript considering the reviewers’ comments. Thank you for the valuable feedback.

Below, you will find the point-to-point revision of the comments.

Reviewer’s comments

Answers to Reviewers

The main concerns pointed out in the last round of review by this reviewer centred on:

1. the performance improvement of RETM wrt ETM, given the added complexity (Figure 5 in previous version)

2. the worse coherence of RETM vs existing models (Figure 6 in previous version).

The authors provide detailed comments or clarifications on these, instead change the presentation of results from in said figures to tables (cf. Tables 2 & 3).

Thank you for the previous comments, we tried to improve readability on the paper so, since one of the graphs was scaled to log2, it might get confusing, we switched to tables (with the real values (not the ones scaled by log2)) and it seems that those are easier to read. Thank you so much!

 Moreover, the presentation of relative performance can be confusing to readers! Is higher or lower perplexity/coherence scores preferred? Review lines 303 - 310: saying method X = more than Y times better, etc. Is this consistent with direction of score preference?

In the results section, when analysing the perplexity metrics, we placed a legend: “where lower is better” and on coherence we added a legend “for topic coherence, more is better.”

About the “X = more than Y times better”, we did a double check, and all are consistent with the direction of score preference.

All the changes were highlighted in blue text color in the document file.

We are looking forward to hearing from you soon.

Best regards,

Carlos Vargas Fraga

Reviewer 3 Report (New Reviewer)

1.The work on LDA and its modifications is very interesting. It's a good tool for working with large databases. I noticed that some improvements have already been implemented. 

2. The methodology could be better expanded; experienced readers will understand it.

3. Of the proposed improvements, have the authors created any new scripts? if so, please provide the link to access them.

4. Have the authors filed a patent? If so, please include it in the text.

5. In the introduction, the authors could add a short paragraph informing readers of the widespread use of LDA. In this way, I recommend for consideration the following applications of LDA in the following papers: https://doi.org/10.1108/JM2-10-2020-0268.

Best regards

Author Response

COVER LETTER

Oct 17th, 2023

Applied Sciences

Dear Editor,

We are pleased to submit our revised manuscript entitled “Recurrent Embedded Topic Model” for possible inclusion in the journal Applied Sciences.

In this revised version, we improved the manuscript considering the reviewers’ comments. Thank you for the valuable feedback.

Below, you will find the point-to-point revision of the comments.

Reviewer’s comments

Answers to Reviewers

The work on LDA and its modifications is very interesting. It's a good tool for working with large databases. I noticed that some improvements have already been implemented. 

Yes, some improvements have been made thanks to your valuable comments, we really appreciate it!

 The methodology could be better expanded; experienced readers will understand it.

Thank you for pointing this out, while reviewing, we noticed that some things were not as clear as expected and thus, we rephrased many things on that section, all these changes can be seen in blue.

Of the proposed improvements, have the authors created any new scripts? if so, please provide the link to access them.

All modifications have already been pushed to the repository of this paper. It can be found at: https://github.com/NxrFesdac/RETM

Have the authors filed a patent? If so, please include it in the text.

There is not patent, the code is free to use.

In the introduction, the authors could add a short paragraph informing readers of the widespread use of LDA. In this way, I recommend for consideration the following applications of LDA in the following papers: https://doi.org/10.1108/JM2-10-2020-0268.

Yes, thank you so much, we added the paragraph with some interesting use cases, one of them, is the paper that you shared us. Thank you!

All the changes were highlighted in blue text color in the document file.

We are looking forward to hearing from you soon.

Best regards,

Carlos Vargas Fraga

This manuscript is a resubmission of an earlier submission. The following is a list of the peer review reports and author responses from that submission.

Round 1

Reviewer 1 Report

The paper attempts to improve the ETM topic model [10] by incorporating the LSTM to enable word order to be considered in modelling topics. Whereas the need to consider word order is important in some applications, the proposed solution is quite straightforward and natural extension of [10]. 

The proposed model, RETM, is thus simple with little novelty. 

The evaluation of the model is centred on the perplexity and PMI metrics. Given the added complexity of RETM (the new RNN architecture), the improvement in performance wrt ETM on perplexity is marginal (as shown in Figure 5). Furthermore, it is not clear (in Figure 6) that the coherence of the proposed model is constantly better than ETM. 

It is difficult based on current empirical evaluation of RETM vs ETM to decide if the extra complexity of RETM leads to significant improvements of the generated topics. Although the authors point to the better training performance of RETM, it is noteworthy that test results do not reflect similarly. Perhaps, ETM is not well trained? Even so, it performs almost just as well as RETM with much better training performance.

Clearly, further evaluation or a good characterisation (or at least an explanation) of the relative performance of RETM vs ETM is required.

Please change "topic modeller(s)" to "topic model(s)" throughout the manuscript. 

Line 57/58 rephrase: "... it still lacks the *possibility of analyze*" to "... it is unable to analyze..."

Rephrase "The LSTM ..." sentence in lines 67 - 69. It is too wordy and difficult to follow.

Reviewer 2 Report

This paper proposes a Recurrent Embedded Topic Model by using the order of the words of the text and the conclusion is that this technique is significantly better than the benchmarking models.

However, the result is only compared with Embedded Topic Modelling and Latent Dirichlet Allocation as baseline approaches and misses the comparison with other similar approaches using Recurrent Embedded Topic Models. 

Also, this paper didn't mention the experiment part of LDA and ETM, which makes the result comparison not convincing as well. 

The Quality of the English Language is good.